



# Age of the Mt. Ortles ice cores, the Tyrolean Iceman and glaciation of the highest summit of South Tyrol since the Northern Hemisphere Climatic Optimum

Paolo Gabrielli[1,2], Carlo Barbante[3,4,5], Giuliano Bertagna[1], Michele Bertó[3], Daniel Binder[6], Alberto Carton[7], Luca Carturan[8], Federico Cazorzi[9], Giulio Cozzi[3,4], Giancarlo Dalla Fontana[8], Mary Davis[1], Fabrizio De Blasi[8], Roberto Dinale[10], Gianfranco Dragà[11], Giuliano Dreossi[3], Daniela Festi[12], Massimo Frezzotti[13], Jacopo Gabrieli[3,4], Stephan P. Galos[14], Patrick Ginot[15,16], Petra Heidenwolf[12], Theo M. Jenk[17], Natalie Kehrwald[18], Donald Kenny[1], Olivier Magand[15,16], Volkmar Mair[19], Vladimir

Mikhalenko[20], Ping Nan Lin[1], Klaus Oeggl[12], Gianni Piffer[21], Mirko Rinaldi[21], Ulrich Schotterer[22], Margit Schwikowski[17], Roberto Seppi[23], Andrea Spolaor[3], Barbara Stenni[3], David Tonidandel[19], Chiara Uglietti[17], Victor Zagorodnov[1], Thomas Zanoner[7] and Piero Zennaro[3]

[1]Byrd Polar and Climate Research Center, The Ohio State University, Columbus, 43210, USA

[2]School of Earth Sciences, The Ohio State University, 275 Mendenhall Laboratory, Columbus, 43210, USA

[3]Department of Environmental Sciences, Informatics and Statistics, Ca' Foscari University of Venice, Venice-Mestre, 30170, Italy

[4]Istituto per la Dinamica dei Processi Ambientali-CNR, Venice-Mestre, 30170, Italy

[5]Accademia Nazionale dei Lincei, Roma, 00196, Italy

[6]Climate Research Section, Central Institute for Meteorology and Geodynamics ZAMG, Vienna, 1190, Austria

[7]Department of Geosciences, University of Padova, Padova, 35131, Italy

[8]Department of Land, Environment, Agriculture and Forestry, University of Padova, Agripolis, Legnaro, 35020, Italy

[9]Dipartimento di Scienze Agro-Alimentari, Ambientali e Animali, Università di Udine, Udine, 33100, Italy

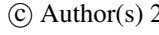



[10]Ufficio Idrografico, Provincia Autonoma di Bolzano, Bolzano, 39100, Italy

[11]Geologin, Varna, 39040, Italy

[12]Institute for Botany, University of Innsbruck, Innsbruck, 6020, Austria

[13]Enea, Roma, 00196, Italy

[14]Institute of Atmospheric and Cryospheric Sciences, University of Innsbruck, Innsbruck, 6020, Austria

[15]Laboratoire de Glaciologie et Géophysique de l'Environnement (LGGE), CNRS, Grenoble, 38041, France

[16]Univ. Grenoble Alpes, Grenoble, 38041, France

[17]Laboratory of Environmental Chemistry, Paul Scherrer Institut, 5232, Villigen PSI, Switzerland

[18]Geosciences and Environmental Change Science Center, U.S. Geological Survey, Denver, 80225, USA

[19]Ufficio Geologia e Prove materiali, Provincia Autonoma di Bolzano, Kardano, 39053, Italy

[20]Institute of Geography, Russian Academy of Sciences, Moscow, 119017, Russia

[21]Waterstones geomonitoring, Egna, 39044, Italy

[22]University of Bern, Bern, 3012, Switzerland

[23]Department of Earth and Environmental Sciences, University of Pavia, Pavia, 27100, Italy

*Correspondence to:* Paolo Gabrielli (gabrielli.1@osu.edu)

**Abstract.** In 2011 four ice cores were extracted from the summit of Alto dell'Ortles (3859 m), the highest glacier of South Tyrol in the Italian Alps. This drilling site is located only 37 km southwest from where the ~5.2 kyrs old Tyrolean Iceman was discovered emerging from the ablating ice field of Tisenjoch (3210 m, near the Italian-Austrian border) in 1991. The

excellent preservation of this mummy suggested that the Tyrolean Iceman was continuously embedded in coeval prehistoric ice and that additional ancient ice was likely preserved elsewhere in South Tyrol. Dating of the ice cores from Alto dell'Ortles based on [210]Pb, [3]H, beta activity and [14]C determinations, combined with an empirical model (COPRA), provides evidence of a chronologically ordered ice stratigraphy from the modern glacier surface down to the bottom ice layers with an age of ~7 kyrs which confirms the hypothesis. Our results indicate that the drilling site was continuously glaciated on frozen

bedrock since ~7 kyrs BP. Absence of older ice on the highest glacier of South Tyrol is consistent with removal of basal ice from bedrock during the Northern Hemisphere Climatic Optimum (6-9 kyrs BP), the warmest interval in the European Alps during the Holocene. Borehole inclinometric measurements of the current glacier flow combined with surface GPR





measurements indicate that, due to the sustained atmospheric warming since the 1980s, an acceleration of the glacier Alto dell'Ortles flow has just recently begun. Given the stratigraphic-chronological continuity of the Mt. Ortles cores over millennia, it can be argued that this behaviour is unprecedented since the Northern Hemisphere Climatic Optimum.

## 5   1 Introduction

Glaciers are sensitive indicators of climate change as their length, area and volume respond primarily to variations in air temperature and precipitation (e.g. Oerlemans, 2001). In general small and steep alpine glaciers show faster response (~10 years) to climate fluctuations than large, less inclined ice bodies (~100 years) (Holzhauser, 1997). In the European Alps, glaciers have undergone major variations at glacial-interglacial time scales as they greatly expanded during the last glacial period and contracted dramatically during the last deglaciation (Ivy-Ochs et al., 2008). While traces of the large expansions of the last glacial period are well preserved even at very low elevations (Ravazzi et al., 2014), evidence of the subsequent smaller Holocene glacier variations is most often overridden by the Little Ice Age expansion (LIA; 14th-19th century). Nevertheless organic fragments (e.g. wood, peat bogs) found recently in forefields of retreating glaciers provide information on the lower altitude limits of past glacial extents, demonstrating that glaciers in the Alps were smaller during the mid-Holocene than they are today (Hormes et al., 2001;Joerin et al., 2008;Joerin et al., 2006;Nicolussi and Patzelt, 2000;Porter and Orombelli, 1985).

The end of the Younger Dryas (11.7 kyrs BP) is generally considered to mark the onset of the Holocene. At that time conditions favourable to glaciers persisted in the European Alps until 10.5 kyrs BP (Ivy-Ochs et al., 2009) when a period of climatic warming started, culminating between 6 and 9 kyrs BP during the so-called Northern Hemisphere Climatic Optimum (Vollweiler et al., 2006). At this time the Northern Hemisphere summer insolation and solar irradiance reached maximum levels during the Holocene (Berger and Loutre, 1991;Stuiver et al., 1998). Specifically, this climatic optimum was characterized by three particularly warm phases at 9.2 kyrs BP, 7.45-6.65 kyrs BP and 6.20-5.65 kyrs BP (Joerin et al., 2008). A climate that was generally unfavourable for glacier advances persisted at least until 6.8 kyrs BP (Luetscher et al., 2011). Between 5.3 and 3.3 kyrs BP the changed climatic conditions marked the beginning of the Neoglaciation in the European Alps (Magny and Haas, 2004;Ivy-Ochs et al., 2009). During this new phase glaciers showed larger variations which culminated in three large LIA expansions (14th, 17th and 19th centuries) (Holzhauser et al., 2005), followed by an on-going phase of intense glacier waning (Zemp et al., 2006).

At the end of summer 1991, the 5200-year old Tyrolean Iceman mummy emerged from the ablating ice field of the Tisenjoch, a saddle at 3210 m near the Italian -Austrian border in the Eastern Alps (Seidler et al., 1992). The excellent state of preservation of the Tyrolean Iceman provides strong evidence for the minimum coverage of this ice field at this time, which has only recently been surpassed. This discovery also suggests that past atmospheric temperatures characterizing warm phases such as the Roman (250 BC – 400 AD) and the Medieval (950-1250 AD) periods may have never exceeded that of the current time in this sector of the Alps (Baroni and Orombelli, 1996). Nevertheless, information about the glaciation of the Eastern Alps before the LIA remains sparse (Nicolussi and Patzelt, 2000).



In this context ice cores can provide useful information. In the Western Alps, an ice core extracted from Colle Gnifetti (4450 m, Monte Rosa, Italian-Swiss border) provided evidence of more than ~10 kyrs old ice in its lower section (Jenk et al., 2009), suggesting a continuous glaciation of at least the highest locations of the western Alps throughout the Holocene. In 1991, at the time of the discovery of the Tyrolean Iceman, only pollen analyses (Bortenschlager et al., 1992)

were performed on the ice in which this mummy was embedded for ~5.2 kyrs at the Tisenjoch, which is now completely deglaciated. This is unfortunate because this now melted ice had the potential to be as old as the Tyrolean Iceman himself and may have preserved a unique snapshot of additional information of the past environmental conditions experienced by the Alpine populations during the mid-Holocene.

In 2010, we formulated the hypothesis that an ice core record encompassing the time of the Tyrolean Iceman was

embedded in the upper reaches of the Alto dell'Ortles (3859 m), the main glacier of Mt. Ortles (3905 m, Italy), which is the highest mountain of South Tyrol, located 37 km from the Tisenjoch (Gabrielli et al., 2010). This idea was based on the following observations: i) Alto dell'Ortles is wind exposed and located in a rain shadow (Schwarb, 2000), and thus is likely characterized by a low accumulation rate compared to the average of the Alps; ii) the upper exposed margins of Alto dell'Ortles show laminated ice layers down to bedrock; and iii) the concomitant significant thickness and moderate tilt (8°-

9°) of the upper part of Alto dell'Ortles may be indicative of minimal basal flow and a frozen ice/bedrock interface, as it could have also been expected at this elevation in this area (Suter et al., 2001).

Until recently, alpine ice core records have been obtained only from the Western Alps (Barbante et al., 2004;Barbante et al., 2001;Jenk et al., 2009;Legrand et al., 2003;Preunkert et al., 2001;Preunkert et al., 2000;Schwikowski et al., 1999b;Schwikowski et al., 1999a;Van de Velde et al., 2000a;Wagenbach et al., 1988) because of their high elevation and

the consequent common occurrence of the cold firn zone (Golubev, 1975;Haeberli and Alean, 1985) which is more likely to retain climatic and environmental signals (Eichler et al., 2001). In contrast, because of their lower elevation, the highest glaciers in the Eastern Alps were assumed to be entirely within the temperate firn zone, and thus unsuitable for preserving intact ice core records (Oerter et al., 1985).

Modern climatic conditions in the Eastern Alps are very unusual because since ~1980 summer air temperatures

have shown a step increase of about 2°C at high elevations (Gabrielli et al., 2010;Auer et al., 2006). Consequently the extensive summer meltwater percolation through the shallow (<10 m) temperate firn layers recently observed on Mt. Ortles could be a relatively recent phenomenon that intensified after ~1980 (Gabrielli et al., 2010). Interestingly, ice which preserved climatic and environmental signals was already found in 2003 below temperate firn on the Quelccaya ice cap in the Peruvian Andes, where recent and strong meltwater percolation did not affect the climatic signal embedded within the

deepest impermeable ice layers (Thompson et al., 2006). We likewise speculated that a climatic signal might still be preserved within the deep ice layers of the glacier Alto dell'Ortles (Gabrielli et al., 2010).

During the Autumn of 2011, working within the framework of an international program aimed at studying past and present climate conditions in the Alps ("Ortles Project", www.ortles.org), we extracted from Alto dell'Ortles the first ice





cores drilled to bedrock in the Eastern Alps (Gabrielli et al., 2012). Here we show that the Alto dell'Ortles cores contain records at millennial time scales and that the bottom ice dates back ~7 kyrs BP. Combining ice core and glaciological observations we discuss these findings in light of the state of the knowledge of the glaciation of the Eastern Alps during the Holocene.

## 2 The drilling site

### 2.1 General characteristics

The Alto dell'Ortles glacier covers the northwestern side of Mt. Ortles, which gently slopes (8º - 9º) from near the summit for ~300 m, then flows on steeper bedrock into two major "tongues" down to 3018 m (Fig. 1). According to some

recent LiDAR measurements (see section 2.2.2), the total surface area was of 1.12 km$^2$ in 2011 and of 1.07 km$^2$ in 2013, of which ~10% constitutes the upper gentle plateau. The ice core drilling campaign was conducted during September and October 2011 on a small col (3859 m; 10°32"34, 46°30"25) between the summit of Mt. Ortles (3905 m) and the Vorgipfel (3845 m) (Fig. 1-2). At the drilling site, the bedrock is at ~75 m of depth (Gabrielli et al., 2012;Gabrielli et al., 2010) and the current accumulation rate (2011-2013) is ~ 800 mm water equivalent (w.e.) per year.

Over the last three decades (1980-2009) the reconstructed average summer (JJA) air temperature was -1.6 °C, ~2 °C higher than during the previous 115 years, with a peak of +2°C during the summer of 2003 (Gabrielli et al., 2010). In 2011, englacial temperatures provided firm evidence of the concomitant presence of a temperate firn portion, deep cold ice layers and a frozen bedrock. In fact, thermistors located within the firn indicated temperatures at or near the pressure melting point while those positioned in the ice (below the firn ice transition at ~30 m depth) clearly demonstrate negative

temperatures at 35 m (-0.4 ºC), 55 m (-1.8 ºC) and at 75 m (-2.8 ºC) close to bedrock, confirming the presence of cold ice (Gabrielli et al., 2012). We concluded that this glacier probably represents a unique remnant of the colder climate prior to ~1980, which has since been shifting from a cold to a temperate state.

### 2.2 Current dynamic

### 2.2.1 Elevation changes

Comparison of terrestrial photographs of Alto dell'Ortles taken in 2010 and during the period from 1900 to 1930 (Fig. 2), suggests a thinning of 8-10 m at the drilling site. Comparison of digital terrain models (DTM) obtained from topographic maps created in 1962 and 1984 (obtained by aerial photogrammetry and provided by the Istituto Geografico Militare and Province of Bolzano, respectively) and from LiDAR surveys in 2005 and 2013 (provided by the Province of

Bolzano and by the Institute of Atmospheric and Cryospheric Sciences, University of Innsbruck (Galos et al., 2015)) indicate a major thinning at the drilling site from 1962 to 1984 (-25±4.7 m). From 1984 to 2005 thickening prevailed (+10.5±7.9 m), followed by minor elevation changes from 2005 to 2013 (-0.7±1.0 m). More extended analyses covering the upper 50 m of



the glacier Alto dell'Ortles, which includes the drilling site, indicate that thinning was widespread from 1962 to 1984 (-9.5±4.7 m) in the upper part of this glacier, followed by rather stationary conditions (-1.5±0.3 m) between 1984 and 2013.

These observations indicate that: i) the upper part of Alto dell'Ortles was subject to significant elevation changes during the last decades; ii) the drilling site itself experienced even larger elevation changes, and; iii) these elevations changes are not directly linked to atmospheric changes (e.g. summer atmospheric warming and surface glacier ablation). It is indeed remarkable that while the site thinned during the relatively cold period between the 1960s and 1980s, most glaciers at lower altitude expanded in this geographic area (Carturan et al., 2013 and references therein). Local elevation changes at the drilling site likely result from the interplay of glacier dynamics and spatial variability of ablation and, notably at this high-elevation site, snow accumulation and redistribution by the wind.

### 2.2.2 Surface and internal dynamics

A borehole displacement of 3.2 m y$^{-1}$ over 1.7 years (5 October 2011-1 July 2013) was determined at the glacier surface by means of differential GPS measurements (Fig. 3). However, the glacial flow was not constant during this period as it varied between 3.7 m y$^{-1}$ (5 October 2011-7 September 2012) and 2.6 m y$^{-1}$ (7 September 2012-1 July 2013), suggesting a seasonal variability characterized by a higher flow in summer than in winter.

The direction of the measured displacement of the boreholes and the variability of the glacial flow during the observed periods are consistent with: i) glacier flow lines originating from the southern flank of Mt. Ortles, whose summit is located 270 m uphill from the drill site (Fig. 3); and ii) basal sliding of the glacier bed lubricated by summer meltwater, perhaps percolated from the outcrops of bedrock located uphill from the drilling site. The surface flow lines have been inferred from a DTM obtained with a LiDAR survey performed during the 2011 ice core drilling campaign, using the flow accumulation tool of ESRI ArcMap™. Importantly, this analysis also shows that the drilling site was located at or in very close proximity to the ice divide (as derived from the surface topography, Fig. 3).

In order to infer the internal dynamics of the glacier, inclinometric measurements were performed in borehole #2 43 days after the end of the drilling operation (5 October -17 November 2011). Further measurements were not possible because of the rupture of the pipe (or the formation of an internal ice lens) at 25 m depth. Uncertainty of this measure is ±6 mm/25 m. A cumulative displacement of 277 mm (2.4 m y$^{-1}$) relative to the bottom part of the inclinometer was observed on the glacier surface (Fig. 4). We also note that the glacier flow decreased linearly with depth, which is inconsistent with the velocity fields typically recorded within glaciers frozen at the bed (Paterson, 1999). While the relative inclinometric measurement does not necessarily imply a net basal sliding of the drilling site, it does indicate that currently ice layers located next to bedrock are dynamically active (38 mm (0.3 m y$^{-1}$) at 65 m; 18 mm (0.2 m y$^{-1}$) at 70 m). This information is important in order to evaluate the age of the basal ice (see discussion in section 6.2).





### 2.3 Bedrock topography

Ground penetrating radar (GPR) was used to determine the bedrock topography (e.g. Binder et al., 2009;Moran et al., 2000) and to infer information about possible englacial features (e.g. Blindow and Thyssen, 1986;Konrad et al., 2013). 50 MHz GPR profiles were collected with a GSSI SIR 3000 system during July 2013. This spatial survey focused on a
region of 50x50 m, including the four 2011 boreholes with an inter-profile distance of 4 m (Fig. 1c). Based on the available continuous snow/firn/ice density data from a snow pit and from borehole #2 (BH2), a 1D-velocity function was derived from the correlation between snow/ice-density and dielectric permittivity by Kovacs et al., 1995. The picked two-way travel times (TWTs) were converted to depth with the 1D velocity function and interpolated to continuous surfaces.

The reflection horizon at TWTs of 800-900 ns (~75-85 m) was interpreted as bedrock. For BH1, BH2, BH3 there is
a good correspondence (within 1 m) between the GPR derived ice thicknesses and the ice core lengths (Fig. 5). The GPR derived ice depth for BH4 indicated that the ice core drilling was stopped ~15 m from bedrock. Since the applied snow/ice density – dielectric permittivity correlation is valid for dry polar conditions, this confirms a dry and cold glacial body at the drilling site. Fig. 5 shows profile slices reverse to the flow direction (X-slices) and orthogonal to the flow direction (Y-slices) through the four boreholes. A continuous internal layer was identified 20-40 m above the bedrock (Fig. 5). The internal layer
could be tracked throughout the investigated area, which suggested an isochronical origin. Two close ice layers at a depth of about 45 m corresponded well with this spatially continuous englacial reflection.

Detailed bedrock topography at the drill site, as obtained by spatial interpolation of GPR point measurements, is illustrated in Fig. 6. We note that the boreholes reached bedrock near a ~10 m step located between the drilling site and the Vorgipfel (Fig. 1,5,6). However, this feature does not completely enclose the drilling site in every direction. We conclude
that while this morphological feature may have facilitated the *in situ* retention of old bottom ice, it is unlikely to have caused a complete dynamical entrapment and the consequent formation of fossil ice decoupled from the upper stratigraphic sequence, which is also implied by the ice flow observed near bedrock (Fig. 4).

### 3 Ice cores characteristics

The four cores were drilled on Alto dell'Ortles within ~10 m (Fig. 1), reaching final logged depths of 73.53 m (#1), 74.88 m (#2), 74.83 m (#3) and 61 m (#4). The drilling of core #4 was stopped for technical issues at a depth ~15 m above bedrock as determined by GPR (see previous section). It has been stored intact for possible future complementary analysis. Likewise, the drilling of core #1 stopped at 73.53 m just short of bedrock because of technical issues. On the other hand, it was obvious that cores #2 and #3 did reach bedrock since further penetration was not possible and damage to the drill cutters
was observed. The comparable final logged depths of core #1, #2 and #3 (73.53, 74.88 m and 74.83 m) are in close agreement with the glacier thickness (75 m) as determined by GPR before (2009) (Gabrielli et al., 2010) and after (2013) the drilling operation (see previous section).

Consistent with the geology of the Mt. Ortles summit, a few limestone rock particles and pebbles were observed in the deepest sections of cores #1, #2 and #3 within ~1 m of bedrock, providing visible evidence of the bed material in the





bottom ice. In core #2 a single large pebble (~1 cm) was observed 2.77 m above bedrock. This was either entrained from the bed or from the glacier surface because of the short distance (~250 meters) of the drilling site from the rock outcrops of the Mt. Ortles summit.

Density measurements of cores #1, #2 and #3 indicate a firn/ice transition at ~30 m depth (Fig 7), with a measured average ice density of 882 kg m$^{-3}$. The air bubbles entrapped within the ice layers throughout the core range from a few mm to less than ~1 mm in diameter. Elongated air bubbles (up to 10-15 mm) are widespread through the cores confirming that flow is a significant component of the ice dynamics of this drilling site (Gabrielli et al., 2012).

Direct observations of the three cores were performed and borehole images were obtained from a 360-degree continuous-imaging scan of borehole #1 (Optical Televiewer, Advanced Logic Technology, Luxemburg). As light reflectance is determined by the concentration and size of air bubbles, this technique highlights the core layers and the presence of ice lenses (melt layers). Horizontal to tilted (10-20°) bubble free ice lenses are present throughout the entire lengths of the cores, where the angles may reflect, at least in part, the tilt of the basal slope. The cumulative thickness of the ice lenses constitutes ~20% of the entire length of the firn portion (when expressed in ice equivalent), ~15% of the glacier ice between 30 and 55 m, and ~5% between 55 and 65 m depth (Gabrielli et al., 2012).

Borehole images of core #1 also suggest strong ice layer thinning from 60-65 m depth to the basal ice (Fig. 7). In this case we interpret the low reflectance of this basal ice as a consequence of the bubbles shrinking due to the overburden pressure over thousands of years (see section 4.3). The digital red index associated with the borehole images (this expresses numerically the red component of each pixel that displays colours as a combination of Red, Green and Blue; RGB color code) suggest that this transition does not occur abruptly (as expected in the case of a physical hiatus) but instead over ~5 m, perhaps indicating a continuous change of the physical properties of the ice over time. This observation is also important in order to evaluate the possible presence of a decoupled fossil ice portion near the bedrock.

## 4 Ice core samples analysis

### 4.1 Fission products

#### 4.1.1 Beta activity and tritium

Depending on the ice mass available, analyses of beta activity (Byrd Polar and Climate Research Center, BPCRC) and tritium (University of Bern and University of Venice) were performed with various degrees of continuity and resolution in various sections of the cores using established methods (Maggi et al., 1998;Schotterer et al., 1998;van der Veen et al., 2001)(Fig 8). Three sections from core #2 (2.75 m, 22.95 m and 41.27 m) were reanalysed for beta activity at the Laboratory of Glaciology and Geophysics of the Environment (LGGE) in Grenoble, France by means of a Berthold LB770-2 gas-flow proportional counter (Ar/CH$_4$ gas) (Magand, 2009;Pourchet et al., 2003;Vimeux et al., 2008), showing variations that are consistent with those determined at BPCRC (Fig. 8).





A well-defined peak of beta and tritium activity can be observed at 41 m depth indicating the 1963 radioactive horizon resulting from nuclear weapon testing, which is consistent with $^{210}$Pb dating (see below). The match between the tritium (entrained in the ice matrix) and beta emission (emitted by ions such as $^{90}$Sr) peaks in core #2 suggests that post-depositional effects due to melt water percolation were negligible from the time of the deposition of this radioactive layer

(1963) until it was entrained below the firn/ice transition. This is also a preliminary indication that the chemical stratigraphy in the firn and in the ice was preserved before the onset of the exceptional current warming (1980 in this area) and the linked surface melting and meltwater percolation (Gabrielli et al., 2012;Gabrielli et al., 2010). Scrutiny of the well-resolved beta record in core #2 indicates a secondary beta emission peak at 44 m depth, likely resulting from the widely reported 1955-1958 thermonuclear tests (Gabrieli et al., 2011).

Evidence of the radioactive fallout from the 9 March 2011 Fukushima nuclear plant accident is observed in the shallow spring-summer 2011 layer (Fig 8). In this case, while tritium levels remain low, beta activity reaches values that are comparable to the residual radioactivity released by the ice layers contaminated by the atmospheric nuclear tests of the 1950s and the beginning of the 1960s. Glaciological evidence of the Fukushima radioactive fallout has already been reported from Tibetan Plateau (Wang et al., 2015) and Arctic (Ezerinskis et al., 2014) snow samples. Our data from the Eastern Alps are

consistent with the clearly detectable Fukushima radioactive fallout widely observed in Europe (Masson et al., 2011), including the northern Italian city of Milan (Clemenza et al., 2012).

An additional peak of beta activity can be observed in the Mt. Ortles cores at ~28 m of depth (Fig 8), near a thick ice lens immediately above the firn ice transition (~30 m). This signal does not seem to be directly related to the 1986 Chernobyl radioactive fallout as its timing is inconsistent with the $^{210}$Pb age determined at this depth (1979.5 ± 3; see section

4.2). It is more likely that percolating summer meltwater transported Chernobyl radionuclides through the porous temperate firn from their actual deposition layers (see next section) down to ~28 m where at least part of the water refroze and the radionuclides were accumulated. This process also suggests that the chemical stratigraphy is not well preserved in the firn portion of the Mt. Ortles cores.

**4.1.2 $^{137}$Cs**

The three sections analysed for beta emissions at LGGE (see previous section) were also analysed by a very low background germanium planar detector at the Laboratoire Souterrain de Modane (LSM – 4800 water equivalent) in France (Loaiza et al., 2011). These measurements were below the detection limit with the exception of the section at 41.27 m (1963), which showed a non-decay corrected activity of 0.035 ± 0.003 (Bq kg$^{-1}$) for $^{137}$Cs. This result is consistent with $^{137}$Cs

values typically linked to the 1963 radioactive deposition (United-Nations-Scientific-Committee-on-the-Effects-of-Atomic-Radiation, 2000).

The undetected $^{137}$Cs in the 22.95 m depth layers (~1986-1987 according to $^{210}$Pb dating, see next section) is notable, taking into account the large $^{137}$Cs quantities released in the European atmosphere during the 1986 Chernobyl



accident (United-Nations-Scientific-Committee-on-the-Effects-of-Atomic-Radiation, 2000). This provides additional support to the idea of the occurrence of post-positional effects due to abundant multiyear meltwater percolation through the firn during the recent warm summers (Gabrielli et al., 2010).

The very low $^{137}$Cs values measured in Mt. Ortles snow within the top 2.75 m (spring-summer 2011) after the
Fukushima accident are consistent with the low levels measured in aerosols over Europe, which is 3 to 4 orders of magnitude lower than activity levels encountered after the Chernobyl event (Mietelski et al., 2014;Povinec et al., 2013). Therefore, it is more likely that the beta activity detected in the Mt. Ortles shallow 2011 snow layer was the product of long-range transport of other Fukushima-derived radionuclides, especially $^{131}$I (Clemenza et al., 2012;Mietelski et al., 2014;Povinec et al., 2013;Lin et al., 2015). Given the short half-life of $^{131}$I (8 days), it might be expected that this layer could be only a short-term
glaciological reference.

### 4.2 $^{210}$Pb

$^{210}$Pb activity was determined continuously in core #2 between 0 and 58.67 m depth at a sample resolution of ~2.80 m length (Table 1) using an established method (Gäggeler et al., 1983;Eichler et al., 2000). The age-depth relationship was
derived from the slope of the linear regression of the logarithmic $^{210}$Pb activity as a function of depth in m w.e. (Fig. 9). The y-axis intercept (109±14 mBq kg$^{-1}$) corresponds to the $^{210}$Pb activity at the surface of the Alto dell'Ortles glacier, and is comparable to values typically observed (~85±10 mBq kg$^{-1}$) in high altitude glaciers in the Alps (Eichler et al., 2000). At a depth of 41.92 m, the calculated age of 54±5 years (1957, 1σ uncertainty) is consistent with the 1963 beta and tritium activity peak found at 41 m (see previous sections). The age of the lowest sample (bottom depth 58.67 m) dated by $^{210}$Pb is
82±7 years (1930).

### 4.3 $^{14}$C analysis

Four large (~1 kg) samples from Mt. Ortles cores #1 (68.96 and 72.48 m) and #3 (71.57 and 74.47 m) (Table 2) were selected for $^{14}$C dating using a recently introduced method based on $^{14}$C determination in the water insoluble organic
carbon fraction (WIOC) of the aerosols in the ice (Jenk et al., 2009;Jenk et al., 2006;Jenk et al., 2007;Sigl et al., 2009). Core #2 was not sampled because sufficient ice volume was not available. Each ice section was divided into three sub-samples (top, middle and bottom), which were filtered separately and analysed. For the section at 72.48 m from core #1, the amount of filtered WIOC from the first sub-sample was estimated to be insufficient and therefore the three sub-samples were filtered together, possibly introducing a larger blank because of the modified treatment (i.e. increased potential for contamination).
$^{14}$C analyses were conducted using the compact radiocarbon AMS system 'MICADAS' at the University of Bern (LARA laboratory). For details about sample preparation, WIOC separation, blank correction and calculation of F$^{14}$C, see (Uglietti et al., 2016). The conventional $^{14}$C ages were calibrated using OxCal v4.2.4 software (Bronk Ramsey and Lee, 2013) with the IntCal13 calibration curve (Reimer et al., 2013). Dates are provided both as cal ages (yr cal BP or simply yrs





BP) and as years before the year of sampling (yr b2011= 2011 (year at surface) – 1950 + yr cal BP). Average values (μ) are presented with a 1 sigma (σ) uncertainty.

The ages obtained from the 3 sub-samples of each section at 68.96 m (core #1) and 71.57 m (core #3) were not significantly different (Table 2) and were therefore combined using OxCal v4.2.4 to obtain single values resulting in average
ages of 590±103 yrs BP and 1609±297 yrs BP, respectively (Table 2). The three subsamples (top, middle, bottom) from the deepest section of core #3 (74.47 m) dated at 4279±619 yrs BP, 5308±633 yrs BP and 6827±425 yrs BP, respectively, indicate very strong glacier thinning close to bedrock and are treated as individual dating horizons. The analysis of the single sample from the section at 72.48 m (core #1) provided an age of 801±135 yrs BP that would represent a chronological inversion with respect to all the other $^{14}$C ages. Considering the above-mentioned increased risk of contamination during
sample preparation this age value was disregarded.

$^{14}$C analyses of a larch leaf found at 73.25 m depth in core #1 was performed by AMS at the National Ocean Sciences Accelerator Mass Spectrometry Facility at the Woods Hole Oceanographic Institutions. This provided an additional, conventionally derived $^{14}$C age of 2612±101 yrs BP, which is stratigraphically and chronologically consistent with the WIOC $^{14}$C ages of 1609±297 yrs BP and 4279±619 yrs BP obtained from core #3 at 71.57 and 74.02 m of depth,
respectively (Table 2). Stratigraphic consistency also remains valid when a common depth scale for the three cores is adopted (see next section).

## 5 Ice core chronology

### 5.1 Depth scale alignment

Cores #1, #2 and #3 were aligned on the same depth scale (Fig. 10) by matching their stable isotope records (smoothed by 3-sample moving averages), which were determined at the BPCRC (cores #2 and #3) and at the University of Venice (#core 1). Core #2 was chosen as the reference core because it was analysed at the highest resolution (3100 samples; 2220 for core #1 and 1038 for core #3) and because it is also the longest core (74.88 m vs. 73.53, 74.83 m for core #1 and #3). Similar features among the δ$^{18}$O records were matched and correlation coefficients (r) were calculated using the
Analyseries 2.0.8 software. Seventeen tie points between core #2 and #1 (r=0.72) and 14 tie points between core #2 and #3 (r=0.67) were used to establish the correspondence between the records. Due to the close proximity of the boreholes, modifications of the original depths of cores #1 and #3 were consistently within 1 meter, which is confirmed by the common stratigraphic control point at ~ 41 m (1963 radioactive peak). The only depth interval in which the profiles of the three cores do not match is near the firn/ice transition (~30 m of depth), probably due to the localized differences in the extent of post-
depositional effects from meltwater percolation. Nevertheless, the generally good correlation between the matching of the three stable isotopic records is compelling evidence of the stratigraphic consistency of the Mt. Ortles cores and further suggests the preservation of climatic and environmental signals.



### 5.2 Depth age relationship

Once a common depth scale for the three cores was established, we developed a continuous depth-age relationship by combining the results obtained from the various independent dating methods applied to the three cores. The chronological references we used are summarized in Table 3. When possible, priority was given to the most accurate time references (e.g.

1963 tritium peak instead of the corresponding [210]Pb determinations). Two approaches were attempted: i) the "Thompson 2-parameter model" (2-p model) (Thompson et al., 2002) successfully applied to an Alpine ice core from Colle Gnifetti to derive an age-depth relationship based on WIOC [14]C dates (Jenk et al., 2009); and ii) an empirical fitting by means of a Monte Carlo simulation (2000 simulation runs) (Breitenbach et al., 2012).

A fit by the 2-p model could not be achieved (not shown) because the degrees of freedom defined by the underlying

simple ice flow model do not account for the observed strong and rapid thinning below ~60 meters of depth (Fig. 7b). On the other hand, the Monte Carlo simulation is purely empirical and does not require a glaciological model of unknown complexity. The resulting age-depth scale is exclusively defined by the reliability and precision of the individual dating horizons, and can account for potential changes in snow accumulation and/or strain rate (Fig. 11). It further allows for an objective uncertainty estimate for each depth, which is defined by the density of dating horizons and their individual

uncertainties. As a consequence, the uncertainty is particularly high (20-50%) between 58 m (1930 AD) and 68 m (1360 AD) where unfortunately no dating horizons exist (Fig. 11).

Future identification of additional dating horizons and counting of annual layers (e.g. using variations in stable isotopes, dust, pollen concentrations and/or species) could help to better constrain this model. However, additional [210]Pb and [14]C techniques cannot be used because the age of the ice is either too high or too low for these methods. Horizons from

volcanic fallout could not be identified so far in the upper and central part of the core, probably due to masking by the significant sulphate deposition from recent anthropogenic activities and the sedimentary background of Mt. Ortles.

## 6 Discussion and implications

### 6.1 Age of the bottom ice

Dating of the ice filtered from the Mt. Ortles cores with the [14]C method employing WIOC provides evidence of a bottom ice age of 6.8±0.4 kyrs BP. The determination of [14]C in a larch leaf in core #1 by traditional methods provides an absolute and accurate timeline (2.6±0.1 kyrs BP) that is chronologically consistent with the [14]C ages of the ice samples (Fig. 11). While the exceptional thinning of the ice layer thicknesses below 58 m is difficult to explain, the monotonicity of the chronological empirical curve (Fig. 11) and of its derivative, together with the gradual change of the red index between 60

and 70 m (Fig. 7) suggests that the thinning process operated consistently, at least at a millennial time scale. Thus the overall ice core stratigraphy is likely continuous over millennia.

Nevertheless, a still unexplained physical process of thinning, or alternatively an unrecognized stratigraphic centennial scale hiatus between 58 and 68 m of depth must have taken place. Similar situations might also have been observed in other non-polar latitude-high altitude glaciers where the ice core records obtained were considered to be





continuous (Herren et al., 2013;Thompson et al., 1995). We note that continuity of the ice core records is relative to the time scale considered as, by their own nature, ice cores are constituted by wet deposition occurring intermittently at different time scales (e.g. meteorological (snow events), seasonal (wet-dry seasons), decadal (droughts caused by recurring patterns of ocean-atmosphere climate variability etc.) and thus stratigraphic hiatuses are the norm rather than the exception. However,

despite the inherent hiatuses, when considering the appropriate time scale (millennial in our case), continuity of the ice core records can be assumed.

The preservation of early Holocene ice in the Alto dell'Ortles glacier is probably due to at least two factors. First, the location of the drill site in a rain shadow and its exposure to wind scour result in relatively low snow accumulation. The medium term snow accumulation at the drilling site was estimated to be 850 mm w.e. /year between 1963 and 2011. This

latter value was obtained by applying the Nye model to correct for firn compaction and ice thinning down to the well-dated 1963 radioactive peak at 41 m of depth; this is consistent with the current 2011-2013 observations of ~800 mm w.e. /year. Second, a frozen bedrock (current value, -2.8 °C) did not allow basal melting and a significant basal flow (see next section), thus preserving in situ the oldest bottom ice.

Last glacial age ice has been observed at several high altitude-low latitude drill sites such as Huascarán (Peruvian

Andes) (Thompson et al., 1995), Sajama and Illimani (Bolivian Andes) (Thompson et al., 1998;Ramirez et al., 2003) and Guliya (Western Tibetan Plateau) (Thompson et al., 1997). In contrast, ice core records from the Western Alps typically extend over just a few centuries (Barbante et al., 2004;Preunkert et al., 2001;Schwikowski et al., 1999a;Van de Velde et al., 2000b). However, the ice embedded in the deepest layers of Alto dell'Ortles dates to the demise of the Northern Hemisphere Climate Optimum (NHCO) and is among the oldest ice discovered in the European Alps, exceeded in age only by the more

than 10 kyrs old ice retrieved at Colle Gnifetti (Jenk et al., 2009). We surmise that last glacial maximum ice does not exist in these two records because of the lack of the significantly stable isotopic depletion (4 to 5‰) characteristic of such ice in the bottom of either Colle Gnifetti or Alto dell'Ortles (Fig. 9). Bottom ice from several other low latitude, high altitude drill sites such as Dasuopu (Himalaya) (Thompson et al., 2000) and Tsambagarav (Western Mongolia) (Herren et al., 2013) behaves in the same way and extends back several millennia to the beginning of the Holocene.

## 6.2 Dynamic of the bottom ice

When considering the location of the drill site in the uppermost part of the glacier Alto dell'Ortles, the current glacier flow (3.2 m/year at the surface, ≥ 0 m/year at the base) and the age of the Mt. Ortles ice cores (6.8 kyrs BP at the bottom), one may wonder why such old ice deposited on the summit of Alto dell'Ortles was not quickly removed during the

Holocene. Here we demonstrate that the only possible answer is that the observed significant ice flow must be a very recent phenomenon, and that consequently a much slower basal flow was typically common since the NHCO.

To study quantitatively the origin of the Mt. Ortles bottom ice and to verify its consistency with the local geography (e.g. the core layers cannot originate from a location beyond the margins of the glacier), we have employed a simple bi-dimensional dynamic model that estimates the *lower limit* of the distance covered by a single glacier layer over

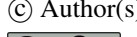


time, under the null hypothesis of an unchanged past dynamic of the glacier from the current conditions. We have therefore assumed a linear variation with depth of the glacier velocity (Vx) (as determined by means of current inclinometric measurements) between a negligible value (0 m/year at the bedrock, current lower limit value) and 2.6 m/year (current lower value recorded at the surface by GPS). For modelling the vertical glacier flow (Vz) we have employed two approaches: i) 5 conservative use of a Nye model, and ii) more realistically, a linear combination of exponential functions that interpolate the empirical chronological timelines obtained along the depth profile.

The results are generally consistent with our ice core sections originating uphill from the drill site along the flow line (Fig. 3). For instance, the 1963 radioactive ice layer (at 41 m) would originate from a location that is at least 90 m uphill of the drill site (using both the conservative and realistic Vz). However, several thousand year old ice (below 70 m of depth) 10 would originate from an unrealistic minimum distance that is 300-500 m uphill (using the conservative and realistic Vz, respectively), thus much larger than the distance between the drill site and the origin of the flow lines (Fig. 3). As this result contradicts the null hypothesis of unchanged dynamic conditions over time, we conclude that the formation and preservation of very old ice at the Alto dell'Ortles drill site was possible only when the flow velocity of the ice layers near the base was much lower. In contrast, the significantly positive glacier flow Vx near the base, which was currently recorded, must be the 15 result of a very recent change in the ice dynamics at the drill site.

We interpret this result as possibly indicative of a large-scale dynamic change, probably involving the entire Alto dell'Ortles. However, this dynamic variation is likely not caused by changes in the slope of this glacier that, according to our large scale comparisons of the DTMs employed (not shown), seem negligible even under the action of the strong ablation occurring since 1980, especially in terms of calving at the lowermost margins of Alto dell'Ortles. Instead, we speculate that 20 this dynamic variation may be a consequence of two possible alternatives or concomitant factors: i) recent summer meltwater influx from the bed outcrops uphill from the basal portion of the drilling site may be lubricating the glacier/bedrock interface. This would be consistent with the observed seasonal changes in the surface velocity, and with the quasi linear profile of Vx obtained with the inclinometer (Fig. 4); ii) changes in the plastic behaviour of the cold portion of the glacier as a consequence of the on-going thermic transition from polythermal to temperate conditions. This latter 25 hypothesis would be consistent with: a) a negligible basal flow, b) long term changes in the vertical thermal profile (perhaps particularly significant since the end of the LIA), and c) the observed significant elevation changes of the drilling site during the last century.

### 6.3 The Alto dell'Ortles glaciation during the Holocene

30 A marked stable isotopic enrichment observed in the speleothem record from Spannagel Cave (2500 m, Austria) during the NHCO (Vollweiler et al., 2006) suggests that this period was likely the warmest during the entire Holocene in the sector of the Alps where Alto dell'Ortles is located. This is consistent with the minimum thickness of the Upper Grindelwald Glacier in Switzerland between 9.2 and 6.8 kyrs BP (Luetscher et al., 2011) and with the minimum extent of the Tschierva





glacier in the adjacent Mt. Bernina group (Joerin et al., 2008), which indicates that the equilibrium line altitude (ELA) was 220±20 m higher than the 1985 reference level (2820 m).

Today the estimated ELA on Alto dell'Ortles lies at about 3300 m as inferred from in situ observations and from the ELA obtained from the nearby glacier of Vedretta della Mare (Carturan, 2016). Under the current warm climatic conditions

the drilling site (3859 m) is polythermal, characterized by temperate firn and cold basal ice. Similarly to the adjacent Mt. Bernina group, the ELA was almost certainly higher than today on Alto dell'Ortles during the NHCO and thus it is certainly possible that this glacier was entirely under a temperate regime throughout its thickness. This would imply the occurrence of basal melting/sliding and thinning of the glacier and the quick removal of bottom ice during the NHCO. This mechanism might explain the absence of ice older than 6.8 kyrs BP at the drill site.

At the end of the NHCO temperatures started to decrease (Vollweiler et al., 2006), probably causing a reversal of the thermal regime of Alto dell'Ortles from temperate to polythermal and thus allowing the accumulation of cold ice on frozen bedrock. As inferred from European paleo-lake levels, a short increase in precipitation at ~7 kyrs BP (Magny, 2004) could also have contributed higher accumulation and ice thickening recorded/inferred at 6.8 kyrs BP for small and climatically sensitive glaciers such as Upper Grindelwald (Luetscher et al., 2011) and Alto dell'Ortles (this work),

respectively. Although high precipitation did not persist during the mid-Holocene, progressively more favourable glacial conditions characterized the Eastern Alps at the end of the NHCO. While there was at least a warm spell (4.4 – 4.2 kyrs BP) in this area (Baroni and Orombelli, 1996), glaciers extended in general to lower elevations, including the Tisenjoch (3210 m) where the Tyrolean Iceman was buried in snow and ice since 5.2 kyrs BP. Today, due to the strong atmospheric summer warming, Alto dell'Ortles is transitioning back from a polythermal to a temperate state. Basal sliding conditions that could

have occurred only during the NHCO are likely to be soon, or are already, fully restored with important and immediate consequences for the dynamic of the entire glacier.

## 7 Conclusions

1) The 2011 Fukushima radioactive signal was embedded in the shallow 2011 snow layers of the Mt. Ortles cores. This

result further suggests that this event may be a new glaciological time horizon at a hemispheric scale, although only in the short-term, due to the short half-life of the radionuclides likely involved such as [131]I.

2) Alto dell'Ortles contains a ~75 m record that spans ~7 kyrs of climatic and environmental history. Dating back to the demise of the Northern Hemisphere Climatic Optimum, Mt. Ortles ice is among the oldest discovered in the European Alps.

3) The Mt. Ortles drilling site was continuously glaciated on frozen bedrock since ~7 kyrs BP. Absence of older ice is

consistent with removal of basal ice from bedrock during the Northern Hemisphere Climatic Optimum.

4) Dating of the upper core layers (mainly performed by means of [210]Pb), and bottom layers (performed by means of [14]C), could only be empirically, but not physically, reconciled. While the Mt. Ortles ice core record can be considered continuous




at a millennial time scale, a centennial hiatus in the stratigraphy between 58 m (1930 AD) and 68 m (1360 AD) cannot be ruled out.

5) During the last century the Mt. Ortles drilling site experienced significant elevation changes (with a net lowering on the order of ~10 m).

6) From 1963 to 2011 the accumulation rate at the drilling site (850 m/year) is comparable to the rate (~800 m/year) measured during the last few warm years (2011-2013).

7) Even assuming negligible basal sliding, detection of a significant shift of the deep ice layers relative to basal ice suggests a recent flow acceleration of the upper portion of Alto dell'Ortles that is unprecedented over ~7 kyrs. Thus we can expect that the old basal ice will be removed long before any possible deglaciation of Mt. Ortles in the future.

**Acknowledgments**

This work is a contribution to the Ortles project, a program supported by two NSF awards # 1060115 & #1461422 to The Ohio State University and by the Ripartizione Protezione antincendi e civile of the Autonomous Province of Bolzano in collaboration with the Ripartizione Opere idrauliche e Ripartizione Foreste of the Autonomous Province of Bolzano and

the Stelvio National Park. This is Ortles project publication 7 and Byrd Polar and Climate Research Center contribution xxxx. The authors are grateful to the alpine guides of the Alpinschule of Solda, the Institute of Mountain Emergency Medicine of EURAC, the helicopter companies Airway, Air Service Center, Star Work Sky and the Hotel Franzenshöhe for the logistical support. We are also grateful for the valuable contribution of Lonnie Thompson in planning/performing the logistic activity and to discuss the ice core data.

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



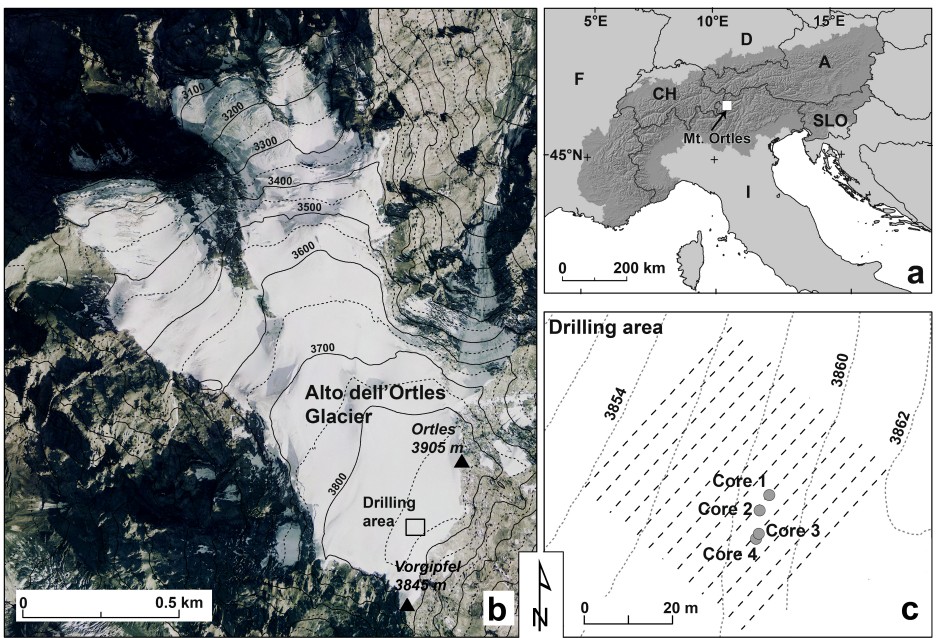

**Figure 1: (a) Geographic location of Mt. Ortles. (b) Map of the Alto dell'Ortles glacier (South Tyrol, Italy), including the area (box) where the drilling operation was conducted during September-October 2011. (c) Detailed map of the drilling site, including i)**
5 **the specific locations where the four cores were extracted and ii) the traces of the ground penetration radar (GPR) survey performed in July 2013.**

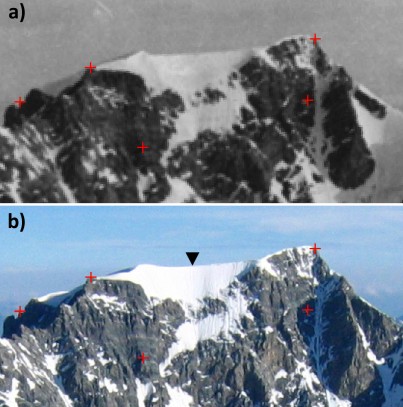

**Figure 2: Comparison of terrestrial photographs of Mt. Ortles taken from the summit of Gran Zebrù (3851 m) a) during the years**
10 **1900-1930 (http://www.montagnedifoto.com/, last access 16 March 2016) and b) on the 4th of July 2010 (photo: Roberto Seppi). The triangle in panel b shows the position of the 2011 drilling site. The four symbols (+) indicate the reference points used for co-registering the two photos (ESRI ArcMap™) before estimating the thickness variation at the drilling site during this period.**



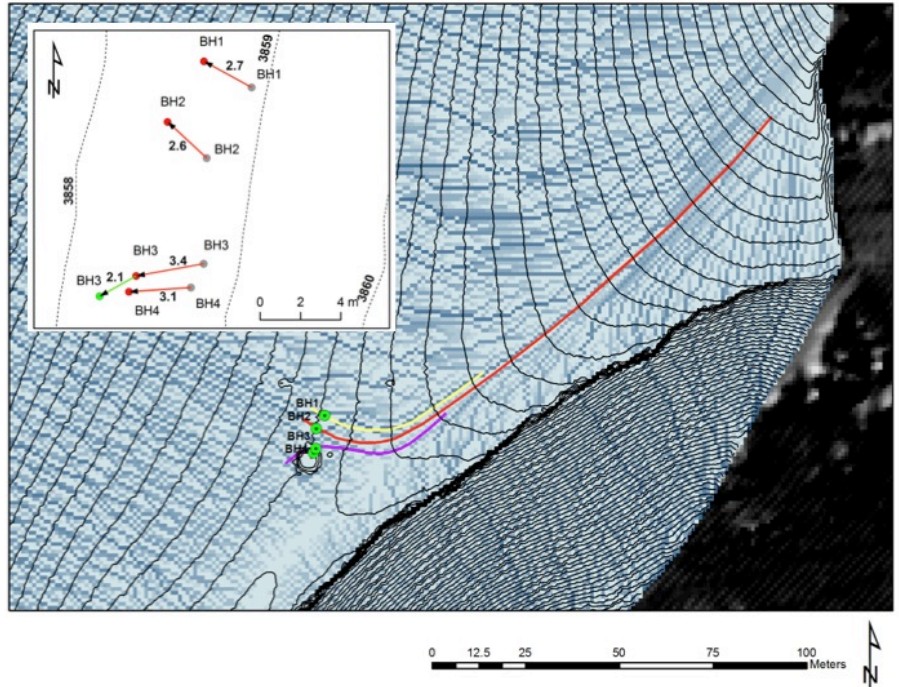

**Figure 3: Reconstructed flow lines and borehole displacements over time. The surface topography of the drill site was obtained from a LiDAR survey conducted during the 2011 campaign (note that the contour lines of the drilling dome are visible). Insert: the displacement between the 5th of October 2011 and the 7th of September 2012 is shown in red while the shift between the 7th of September 2012 and the 1st of July 2013 is in green (borehole #3, only). The values indicate the displacement (in meters) measured by GPS during these two periods.**

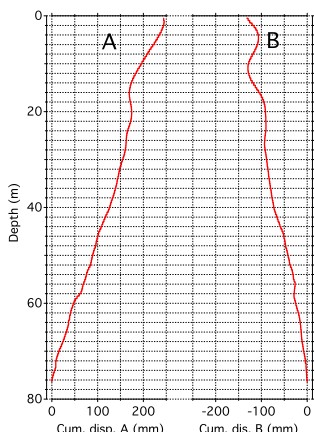

**Figure 4: Relative cumulative displacement of borehole #2 along the two axes (A axis, 340° N; B axis, 70° N) during the 43 days after the end of the 2011 drilling operation. The cumulative displacement is relative as it uses as a reference the bottom portion of the inclinometer (not the bedrock).**





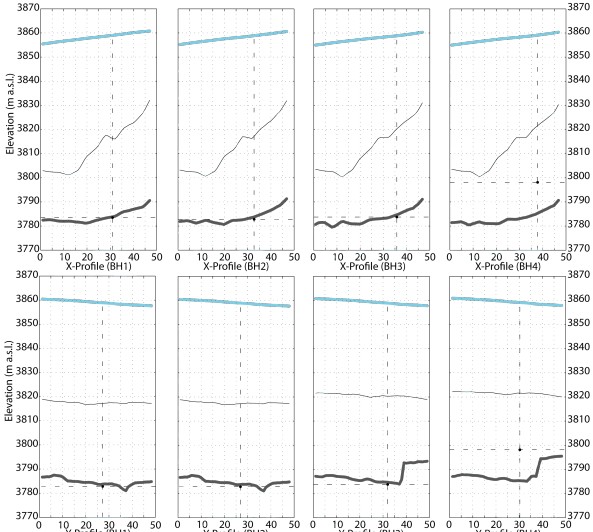

**Figure 5: The top images illustrate the X-profiles (constant x-coordinate) and Y-profiles (constant y-coordinate) through the locations of boreholes #1-4 (see Fig. 1). Two-dimensional slices of the interpolated bedrock surfaces (thick grey line) and a possible englacial layer (thin grey line) are shown. The thick blue line indicates the glacier surface. Black dots show the logged depths for boreholes #1-4.**

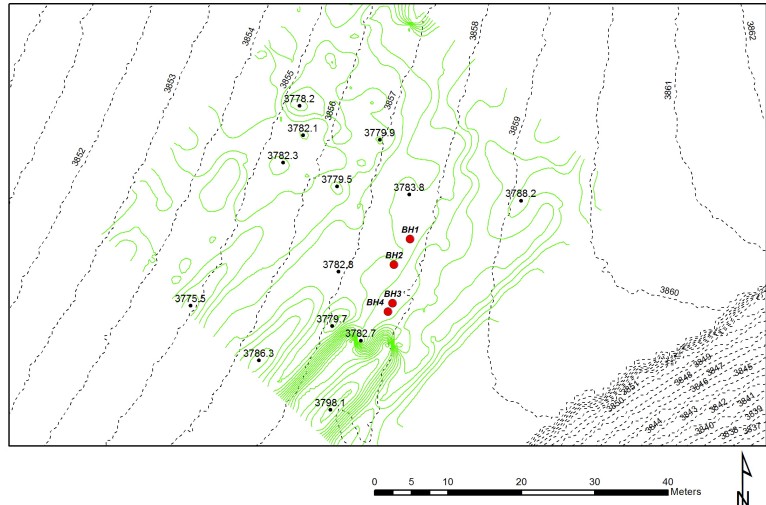

**Figure 6: Bedrock contours and surface topography of the 2011 ice core drill site. Data were obtained during the 2013 GPR and lidar surveys, respectively. Bedrock contours are shown in green while the surface topography is displayed in black. The positions where the four cores were extracted in 2011 are shown as filled red circles.**





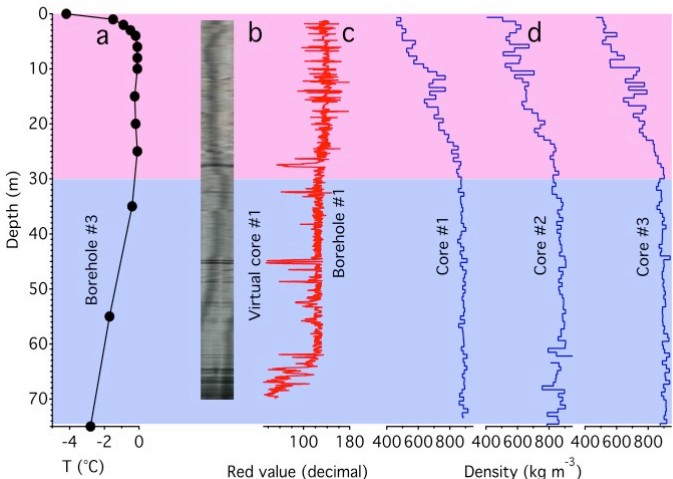

**Figure 7: Physical characteristics of the Mt. Ortles cores. The temperate firn portion is enclosed in red shading while the cold ice is in blue. a) Borehole #3 temperatures recorded 43 days after the end of the drilling operations (from Gabrielli et al., 2012) b) Virtual image of core #1 reconstructed from 360° Televiewer visual scanning of borehole #1. c) Red component of the RGB digital**
5 **signal obtained by means of visual scanning. High values indicate higher light reflection. d) Densities of the Mt. Ortles ice cores #1, #2 and #3.**

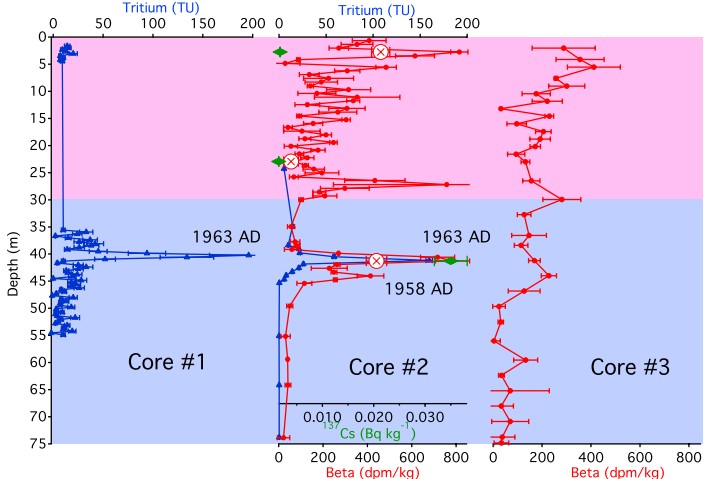

10 **Figure 8: Fission products determined in the Mt. Ortles cores #1, #2 and #3. The temperate firn portion is enclosed in red shading while the cold ice is in blue. Tritium is depicted by blue triangles, beta emissions by red dots (BPCRC) or red crosses (LGGE) and ¹³⁷Cs by green diamonds. Beta emissions in cores #2 and #3 are determined at different resolutions depending on the available sample mass.**




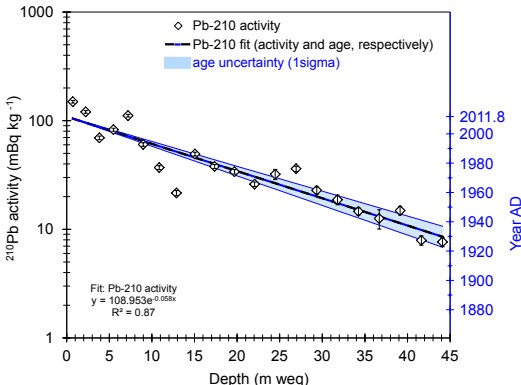

**Figure 9: Measured $^{210}$Pb activity (left y axis, logarithmic scale) and calculated age (right y axis) vs. depth (in meters of water equivalent) relationship in the Mt. Ortles core #2. The best linear fit is also reported for $^{210}$Pb activity (in black) and age (in blue), the latter including 1 sigma uncertainty (in blue).**

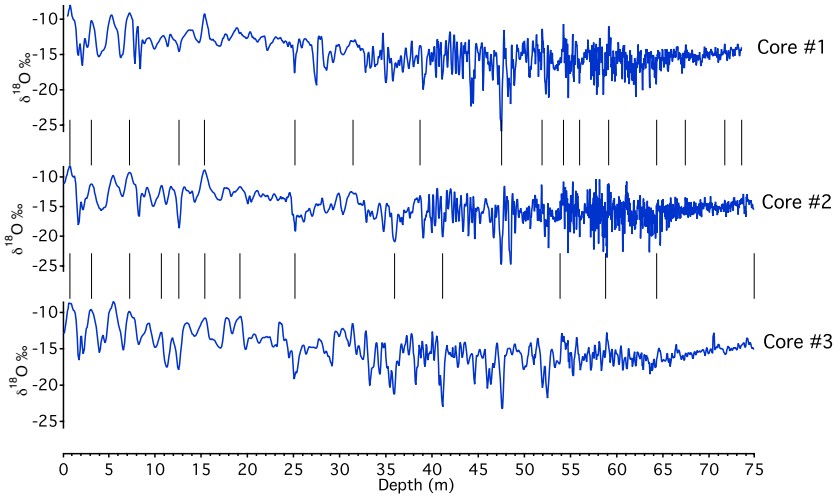

**Figure 10: Alignment of the stable isotopic profiles from Mt. Ortles cores #1, #2 and #3 by depth. Vertical bars indicate the points used to tie cores #1 and #3 to core #2. The latter is used as the reference.**



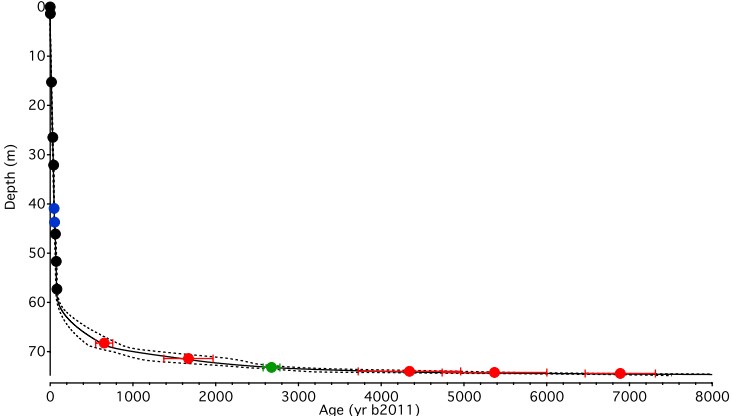

**Figure 11: Age-depth relationship in the Mt. Ortles cores based on 2000 Monte Carlo realizations (continuous black line; dotted line refers to the linked time uncertainty: 2.5% and 97.5% quantile) fitting the absolute empirical time lines used to construct the time scale. Surface constrained and $^{210}$Pb ages (black dots); tritium 1963 and beta 1958 peaks (blue dots); $^{14}$C WIOC determinations (red dots); $^{14}$C determination in the larch leaf (green dot). The empirical values are reported with 1 sigma uncertainty (see also Table 3).**

| Top Depth (m) | Bottom depth (m) | Top depth (m w.e.) | Bottom depth (m w.e.) | Activity in 2011 (mBq/kg) | Uncertainty (mBq/kg) | Age (yr b2011) | Lower age (yr b2011, 1σ) | Upper age (yr b2011, 1σ) | Age (AD) |
|---|---|---|---|---|---|---|---|---|---|
| 0.00 | 2.75 | 0.00 | 1.46 | 149.4 | 3.9 | 1.4 | 1.2 | 1.5 | 2010.4 |
| 2.75 | 5.56 | 1.46 | 3.02 | 120.4 | 3.4 | 4.2 | 3.8 | 4.5 | 2007.6 |
| 5.56 | 8.31 | 3.02 | 4.66 | 69.5 | 2.6 | 7.1 | 6.5 | 7.8 | 2004.7 |
| 8.31 | 11.09 | 4.66 | 6.33 | 82.7 | 2.7 | 10.2 | 9.3 | 11.1 | 2001.6 |
| 11.09 | 13.85 | 6.33 | 8.08 | 111.3 | 3.1 | 13.4 | 12.2 | 14.6 | 1998.4 |
| 13.85 | 16.64 | 8.08 | 9.89 | 60.4 | 2.2 | 16.7 | 15.2 | 18.2 | 1995.1 |
| 16.64 | 19.44 | 9.89 | 11.84 | 37.0 | 1.5 | 20.2 | 18.4 | 22.0 | 1991.6 |
| 19.44 | 22.25 | 11.84 | 13.92 | 21.6 | 1.2 | 23.9 | 21.8 | 26.0 | 1987.9 |
| 22.25 | 25.05 | 13.92 | 16.20 | 49.3 | 2.1 | 28.0 | 25.5 | 30.4 | 1983.8 |
| 25.05 | 27.88 | 16.20 | 18.52 | 37.8 | 1.7 | 32.3 | 29.4 | 35.1 | 1979.5 |
| 27.88 | 30.68 | 18.52 | 20.88 | 33.9 | 1.6 | 36.6 | 33.4 | 39.8 | 1975.2 |
| 30.68 | 33.52 | 20.88 | 23.24 | 26.0 | 1.3 | 41.0 | 37.4 | 44.6 | 1970.8 |
| 33.52 | 36.34 | 23.24 | 25.71 | 32.2 | 3.5 | 45.5 | 41.5 | 49.5 | 1966.3 |
| 36.34 | 39.15 | 25.71 | 28.15 | 36.1 | 2.0 | 50.0 | 45.7 | 54.4 | 1961.8 |
| 39.15 | 41.92 | 28.15 | 30.54 | 22.8 | 1.7 | 54.5 | 49.8 | 59.3 | 1957.3 |
| 41.92 | 44.74 | 30.54 | 33.03 | 18.7 | 1.9 | 59.1 | 53.9 | 64.2 | 1952.7 |
| 44.74 | 47.48 | 33.03 | 35.47 | 14.6 | 1.1 | 63.7 | 58.1 | 69.2 | 1948.1 |
| 47.48 | 50.27 | 35.47 | 37.90 | 12.6 | 2.5 | 68.2 | 62.2 | 74.1 | 1943.6 |
| 50.27 | 53.07 | 37.90 | 40.37 | 14.9 | 1.3 | 72.7 | 66.4 | 79.1 | 1939.1 |
| 53.07 | 55.88 | 40.37 | 42.88 | 7.9 | 0.8 | 77.4 | 70.6 | 84.1 | 1934.4 |
| 55.88 | 58.67 | 42.88 | 45.35 | 7.6 | 0.7 | 82.0 | 74.8 | 89.1 | 1929.8 |

**Table 1: Determinations of $^{210}$Pb activity and calculated ages, by means of the best fit linear regression, in the upper part of the Mt. Ortles ice core #2 (0-58.67 m). Note the used notation for ages in yr b2011, which is defined as the year before sampling (2011).**



| Core # | Tube # | Measure | Top depth (m) | Bottom depth (m) | WIOC (µg) | fM | $^{14}$C age (yr BP) | Cal age (yr BP) | µ cal age (yr BP)** | µ cal age (yr b2011)** | σ (yr)** |
|---|---|---|---|---|---|---|---|---|---|---|---|
| 1 | 98b | WIOC | 68.26 | 68.49 | 17.11 | 0.97 ± 0.03 | 212 ± 220 | (1950 - 428) | 277 | 339 | 172 |
| 1 | 98b | WIOC | 68.49 | 68.73 | 17.86 | 0.91 ± 0.02 | 737 ± 209 | (538 - 906) | 725 | 787 | 189 |
| 1 | 98b | WIOC | 68.73 | 68.96 | 15.06 | 0.90 ± 0.03 | 830 ± 242 | (552 - 976) | 815 (590) | 877 (652) | 219 (103) |
| 3 | 102 | WIOC | 70.87 | 71.14 | 7.98 | 0.78 ± 0.05 | 1971 ± 536 | (1372 - 2697) | 2065 | 2127 | 616 |
| 3 | 102 | WIOC | 71.14 | 71.35 | 7.15 | 0.87 ± 0.07 | 1125 ± 693 | (500 - 1865) | 1299 | 1361 | 713 |
| 3 | 102 | WIOC | 71.35 | 71.57 | 13.28 | 0.82 ± 0.04 | 1616 ± 349 | (1181 - 1948) | 1605 (1609) | 1667 (1671) | 387 (297) |
| 1 | 103b | WIOC | 71.8 | 72.48 | 10.37 | 0.90 ± 0.02 | 841 ± 154 | (670 - 920) | 801 | 863 | 135 |
| 1 | 105b | Larch leaf | 73.25 | 73.25 | 68* | 0.728 ± 0.006 | 2550 ± 65 | (2500 - 2752) | 2612 | 2674 | 101 |
| 3 | 106 | WIOC | 73.73 | 74.02 | 10.91 | 0.62 ± 0.04 | 3806 ± 476 | (3608 - 4846) | 4279 | 4341 | 619 |
| 3 | 106 | WIOC | 74.02 | 74.24 | 11.50 | 0.56 ± 0.04 | 4649 ± 512 | (4629 - 5934) | 5308 | 5370 | 633 |
| 3 | 106 | WIOC | 74.24 | 74.47 | 18.47 | 0.48 ± 0.02 | 5965 ± 403 | (6406 - 7262) | 6827 | 6889 | 425 |

*Pure C extracted after combustion
** In parenthesis, combined values from the three sub-samples of tubes 98b and 102

**Table 2:** $^{14}$C analyses of the particle organic fraction (WIOC) obtained from the four sections (tubes) of the Mt. Ortles ice cores #1 and #3. Except for section 103b, the samples were analysed in three sub-samples (top, middle, bottom). $^{14}$C determination in section 105b (core #1) refers to a larch leaf that was found in the ice. Note the notation used for calibrated ages in yr b2011, which is defined as the year before sampling.

| Time reference | Top depth (m)* | Bottom depth (m)* | Mid depth (m)* | Age (yr b2011) | Age (yr AD, BC) | σ age (yrs) |
|---|---|---|---|---|---|---|
| Surface constrain | 0.00 | 0.00 | 0.00 | 0 | 2011.8 | 0 |
| $^{210}$Pb | 0.00 | 2.75 | 1.38 | 1.4 | 2010.4 | 0.1 |
| $^{210}$Pb | 13.85 | 16.64 | 15.25 | 16.7 | 1995.1 | 1.5 |
| $^{210}$Pb | 25.05 | 27.88 | 26.47 | 32.3 | 1979.5 | 2.8 |
| $^{210}$Pb | 30.68 | 33.52 | 32.10 | 41.0 | 1970.8 | 3.6 |
| Tritium peak | 40.58 | 41.27 | 40.92 | 48 | 1963 | 1 |
| Beta emission peak | 43.33 | 44.04 | 43.69 | 53 | 1958 | 1 |
| $^{210}$Pb | 44.74 | 47.48 | 46.11 | 63.7 | 1948.1 | 5.6 |
| $^{210}$Pb | 50.27 | 53.07 | 51.67 | 72.7 | 1939.1 | 6.4 |
| $^{210}$Pb | 55.88 | 58.67 | 57.28 | 82.0 | 1929.8 | 7.2 |
| $^{14}$C in WIOC | 67.90 | 68.61 | 68.26 | 652 | 1360 | 103 |
| $^{14}$C in WIOC | 71.15 | 71.60 | 71.38 | 1671 | 341 | 297 |
| $^{14}$C in larch leaf | 73.19 | 73.19 | 73.19 | 2674 | -662 | 101 |
| $^{14}$C in WIOC | 73.84 | 74.11 | 73.98 | 4341 | -2329 | 619 |
| $^{14}$C in WIOC | 74.11 | 74.31 | 74.21 | 5370 | -3358 | 633 |
| $^{14}$C in WIOC | 74.31 | 74.53 | 74.42 | 6889 | -4877 | 425 |

*All depths referred to the Ortles core #2 depth scale

**Table 3: Data used in the depth-age modelling.**