# Peer review of "Age of the Mt. Ortles ice cores, the Tyrolean Iceman and glaciation of the highest summit of South Tyrol since the Northern Hemisphere Climatic Optimum"

_The Cryosphere, 2016_

## Short Comment (SC1) · 17 Jul 2016

The discovery of mid Holocene basal ice following the mid Holocene climate optimum is an important observation. It is convincing that the ice at the study location has apparently never fully melted, despite several late Holocene warm phases (e.g. Roman and Medieval Warm Periods).

While the manuscript discusses possible reasons why the ice might have survived at the study location, it should be made clearer, in my opinion, that the "continuous" ice coverage in the region might in fact be more an exception rather than a rule. The

location of the ice core in a rain shadow zone etc. may have helped. I therefore find the last sentence of the abstract too strong:

"Given the stratigraphic-chronological continuity of the Mt. Ortles cores over millennia, it can be argued that this behaviour is unprecedented since the Northern Hemisphere Climatic Optimum."

"Unprecedented" yes, but proven only for this (and the Iceman) locations, and not likely to be representative for the entire Eastern Alps area. Therefore better: "...that this behaviour is unprecedented AT THIS LOCATION since the..."

Likewise, another statement on page 3 (lines 31-33) needs additional context information, in order to avoid over-generalization:

"This discovery also suggests that past atmospheric temperatures characterizing warm phases such as the Roman (250 BC – 400 AD) and the Medieval (950-1250 AD) periods may have never exceeded that of the current time in this sector of the Alps (Baroni and Orombelli, 1996)."

The cited reference from 1996 is now 20 years old. Meanwhile more recent studies have demonstrated that the proposed concept is too simplistic. At Lake Silvaplana in the Upper Engadine (which is located only 65 km west of the ice core), Larocque-Tobler et al 2010c based on chironomids documented that the Medieval Warm Period (MWP) from 1030 AD (start of dataset) until 1260 AD was up to 1°C warmer than the modern climate reference period (1961–1990). http://www.sciencedirect.com/science/article/pii/S0277379110001277

The warm MWP in nearby case studies provides additional evidence that mid to late Holocene ice preservation may have been rather patchy than ubiquitious in the Eastern Alps. It may be worth clarifying and discussing this point in the final manuscript.
* * *

---

## Referee Comment (RC1) · Anonymous Referee #1 · 1 Aug 2016

The manuscript (MS) presents an age scale of three $\sim$75m deep ice cores drilled from near the summit of Alto dell'Ortles glacier in the Italian Alps. The MS is generally well written and argues convincingly for the suggested age scale that is based on radiometric ages and nuclear fallout products. The deepest meters of the ice cores contain ice that is more than 1000 years old, and the MS argues that the glacier was formed during the Northern Hemisphere Climatic Optimum (NHCO). I have a few suggestions for the authors to consider:

In Figure 5, the orientation of the coordinate system is unclear to me. Are the X and

Y directions following the GPR lines in Figure 1c or are they respectively along and perpendicular to the ice flow as stated in the main text? The best would probably be to show the orientation of the X-profiles and Y-profiles in a map. Are the units on the abscissas of Figure 5 meters?

In section 6.2, the authors argue that the only way the obtained age profile can come about is if the glacier flow pattern has recently changed significantly. I have difficulties following the argumentation of this section as not many details are given. A simple model is applied, but no results are presented. There is one observation in Figure 5 that seems important to me in this context. The inclination of the 45 m deep melt layer seen in the GPR profile and in the cores is very steep. This isochrone suggests that the oldest ice of the glacier probably is to be found in right hand side of figures 5 (X-profiles), which is under the ice divide, if I got the geometry right. Alternatively, it suggests that the oldest ice is to be found in higher depth resolution below the ice divide. I'm uncertain if the authors take this observation into account, but to me the steep inclination of the melt layer suggests a significant increase in snow accumulation the further one gets away from the ice divide. Could it be that snow is blown away from the ice ridge and (re-)deposited further down the slope (on the lee side?). An increasing accumulation away from the ice ridge would probably lead to strong inclination of the deeper layers of the glacier and possibly explain why old ice is preserved close to the ice ridge where accumulation may be very low?

In section 3, it is mentioned that limestone rock particles and pebbles are observed in the lowest meter of the ice cores. At the same time, the oldest ice is found in stratigraphic order in the same lowest meter of the cores. Indeed, the proposed age scale does look convincing based on the obtained C-14 ages, but still I am wondering how those pebbles got entrained in the ice if the ice is not disturbed (folded)?

In section 2.2.2 it is suggested that the glacier bed could be lubricated by summer meltwater. This scenario seems rather implausible to me. If the ice is -2.8 C at 75 m depth, it seems highly unlikely that there is summer melt. There is no seasonal

temperature variability possible at this depth.

In section 2.3 it is mentioned about the bedrock step that 'this feature does not completely enclose the drilling site in every direction'. In fact, the bump is only to one side of the cores, so I would suggest a reformulation.

In Figure 3 it is hard to see details around the drilling sites in the main image. There is a white spot right below what appears to be BH3 and BH4. Is this a col, or does the spot indicate something else?

The authors argue that the ice started to form during the Holocene climatic optimum and that this same optimum is observed in the isotopic signature of nearby stalagmites. Is the NHCO seen in the isotopic signature of the deepest ice core ice? If the high frequency signal is removed? I cannot judge this from Figure 10.

In Tables 2 and 3 it is difficult to see which samples correspond to each other as depth is transferred between the cores. Would it be possible to assign a unique name to each sample, so the reader can trace them from one table to the other?

The conclusion seems to state the main findings in a bit disorganized way. In point 6) is says the accumulation at the drilling site is 850 m/year. Probably this should be 850 mm/year?

---

## Referee Comment (RC2) · M. Kuhn (Referee) · 13 Sep 2016

Michael Kuhn michael.kuhn@uibk.ac.at

The glacier Alto dell' Ortles is an exceptional site for drilling into the Holocene history in the Eastern Alps. Its value and uniqueness were recognized in the Ortles Project (www.ortles.org) which produced three ice cores at 3859 m elevation that go back to

about 7000 years and were analyzed by a large team of authors and laboratories who ensured the competent representation of the various disciplines involved in paleoclimate research. Analysis of ice cores requires them to be cold or polythermal, to have minimum annual accumulation and to have a flat topography that reduces the outflow of ice and conserves it in place for millennia. These conditions are not met by many mountains in the Eastern Alps. Suter and others (2001) modelled the firn temperature in the Alps and found from altitude and exposition that cold firn should exist above 3400 m in northerly aspect and above 4150 on south slopes. In the Eastern Alps this would include the peaks of Disgrazia, Bernina, Adamello, Ortles, Cevedale, Ötztal, Stubai, Zillertal, Venediger and Goßglockner. The second condition, that there should be flat glacier tops, ruled out most of these mountains. Topographically suitable candidates like Adamello at 3539 or Weißseespitze at 3510 m in the Ötztal Alps, however, turned temperate in the 1990s. Other peaks like Wildspitze 3768 m, and Venediger 3666 m may still be cold but their steep slopes do not keep ice in place for long. Alto dell' Ortles was timely chosen as research site. The lowermost meters of the Ortles ice cores are difficult to interpret, as is the slope of the internal layers detected by ground penetrating radar. They may reflect wind drift of snow, ablation or ice flow, all of which may have individual histories on a centennial time scale and may act on a very local scale as indicated by the bedrock contours in Fig. 6. As the paper refers to the Tyrolean Iceman – he too was found in a shallow bedrock depression, protected from shearing ice motion. But shear must have occurred above at a vertical distance comparable to the horizontal scale of bedrock roughness, and such shear could be one explanation of a possible hiatus in ice core layering. This paper is of high current interest and deserves publication with minor changes as suggested by two previous comments. I recommend numbering the profiles in Fig. 5 and entering them into Fig. 6. On page 2 / line 2 write Geologist instead of Geologin. 2/4: ENEA instead of Enea. 2/10 Prove Materiali. Best wishes, Michael Kuhn

Sutter S, Laternser M, Haeberli W, Frauenfelder R, Hoelzle M 2001 Cold firn and ice of high-altitude glaciers in the Alps: measurements and distribution modelling. J Glac 47

(156) 85-96.

---

## Author Comment (AC3) · 13 Oct 2016

Authors: Thank you very much for your review and your comments.

M. Kuhn: The glacier Alto dell' Ortles is an exceptional site for drilling into the Holocene history in the Eastern Alps. Its value and uniqueness were recognized in the Ortles Project (www.ortles.org) which produced three ice cores at 3859 m elevation that go back to about 7000 years and were analyzed by a large team of authors and laboratories who ensured the competent representation of the various disciplines involved in paleoclimate research. Analysis of ice cores requires them to be cold or polythermal,

to have minimum annual accumulation and to have a flat topography that reduces the outflow of ice and conserves it in place for millennia. These conditions are not met by many mountains in the Eastern Alps. Suter and others (2001) modelled the firn temperature in the Alps and found from altitude and exposition that cold firn should exist above 3400 m in northerly aspect and above 4150 on south slopes.

Authors: we have added this last sentence and have provided more emphasis to the Suter et al. 2001 reference in the introduction.

M. Kuhn: In the Eastern Alps this would include the peaks of Disgrazia, Bernina, Adamello, Ortles, Cevedale, OÌĹtztal, Stubai, Zillertal, Venediger and Goßglockner. The second condition, that there should be flat glacier tops, ruled out most of these mountains. Topographically suitable candidates like Adamello at 3539 or Weißseespitze at 3510 m in the OÌĹtztal Alps, however, turned temperate in the 1990s. Other peaks like Wildspitze 3768 m, and Venediger 3666 m may still be cold but their steep slopes do not keep ice in place for long. Alto dell' Ortles was timely chosen as research site. The lowermost meters of the Ortles ice cores are difficult to interpret, as is the slope of the internal layers detected by ground penetrating radar. They may reflect wind drift of snow, ablation or ice flow, all of which may have individual histories on a centennial time scale and may act on a very local scale as indicated by the bedrock contours in Fig. 6. As the paper refers to the Tyrolean Iceman – he too was found in a shallow bedrock depression, protected from shearing ice motion. But shear must have occurred above at a vertical distance comparable to the horizontal scale of bedrock roughness, and such shear could be one explanation of a possible hiatus in ice core layering.

Authors: we believe that while the depression may have facilitated the in situ retention of old bottom ice, it is unlikely to have caused a complete dynamical entrapment and the consequent formation of fossil ice decoupled from the upper stratigraphic sequence, which is also implied by the ice flow observed near bedrock.

M. Kuhn: This paper is of high current interest and deserves publication with minor changes as suggested by two previous comments.

M. Kuhn: I recommend numbering the profiles in Fig. 5 and entering them into Fig. 6.

Authors: Done.

M. Kuhn: On page 2 / line 2 write Geologist instead of Geologin.

Authors: Geologin is the name of the company.

M. Kuhn: 2/4: ENEA instead of Enea.

Authors: Done. Thank you.

M. Kuhn: 2/10 Prove Materiali. Best wishes, Michael Kuhn

Sutter S, Laternser M, Haeberli W, Frauenfelder R, Hoelzle M 2001 Cold firn and ice of high-altitude glaciers in the Alps: measurements and distribution modelling. J Glac 47 (156) 85-96.

---

## Author Response (AR1)

Columbus, October the 17th, 2016

Dear Professor M. van den Broeke,

I'm pleased to submit the revised version of the manuscript (MS# tc-2016-159) for your kind consideration. The title is detailed below.

" Age of the Mt. Ortles ice cores, the Tyrolean Iceman and glaciation of the highest summit of South Tyrol since the
10   Northern Hemisphere Climatic Optimum"

We very much appreciate the useful and constructive comments submitted by the referees. We carefully considered the comments point by point when preparing this revised version (please, see below). In the revised manuscript we believe that we have successfully addressed the few concerns and improved our manuscript with other minor modifications, as detailed
15   below.

The corresponding author is:

20   Dr. Paolo Gabrielli
Ice Core Paleoclimatology Research Group
Byrd Polar and Climate Research Center
The Ohio State University
108 Scott Hall, 1090 Carmack Road
25   Columbus, OH 43210-1002
USA
Tel: +614 2926664
Fax: +614 2924697
E-mail: gabrielli.1@osu.edu

Sincerely Yours

Paolo Gabrielli

Sebastian Luening

Authors: Thank you very much for your interest in our work and your comments.

5  Sebastian Luening : The discovery of mid Holocene basal ice following the mid Holocene climate optimum is an important observation. It is convincing that the ice at the study location has apparently never fully melted, despite several late Holocene warm phases (e.g. Roman and Medieval Warm Periods).

While the manuscript discusses possible reasons why the ice might have survived at the study location, it should be made clearer, in my opinion, that the "continuous" ice coverage in the region might in fact be more an exception rather than a rule.

10  The location of the ice core in a rain shadow zone etc. may have helped. I therefore find the last sentence of the abstract too strong:

"Given the stratigraphic-chronological continuity of the Mt. Ortles cores over millennia, it can be argued that this behaviour is unprecedented since the Northern Hemisphere Climatic Optimum."

"Unprecedented" yes, but proven only for this (and the Iceman) locations, and not likely to be representative for the entire

15  Eastern Alps area. Therefore better: "...that this behaviour is unprecedented AT THIS LOCATION since the..."

Authors: the text has been changed accordingly (page 3- line 3).

Sebastian Luening: Likewise, another statement on page 3 (lines 31-33) needs additional context information, in order to

20  avoid over-generalization:

"This discovery also suggests that past atmospheric temperatures characterizing warm phases such as the Roman (250 BC – 400 AD) and the Medieval (950-1250 AD) periods may have never exceeded that of the current time in this sector of the Alps (Baroni and Orombelli, 1996)."

The cited reference from 1996 is now 20 years old. Meanwhile more recent studies have demonstrated that the proposed

25  concept is too simplistic. At Lake Silvaplana in the Upper Engadine (which is located only 65 km west of the ice core), Larocque-Tobler et al 2010c based on chironomids documented that the Medieval Warm Period (MWP) from 1030 AD (start of dataset) until 1260 AD was up to 1°C warmer than the modern climate reference period (1961–1990). http://www.sciencedirect.com/science/article/pii/S0277379110001277

The warm MWP in nearby case studies provides additional evidence that mid to late Holocene ice preservation may have

30  been rather patchy than ubiquitious in the Eastern Alps. It may be worth clarifying and discussing this point in the final manuscript.

Authors: we have included this reference within the text (3-35) and noted that, according to Larocque-Tobler et al 2010, the Medieval Warm Period and modern temperatures would be 1°C warmer than the modern climate reference period (1961–1990) in the Eastern Alps.

Anonymous Referee #1

Authors: Thank you very much for your review and your comments.

Referee #1: The manuscript (MS) presents an age scale of three    75m deep ice cores drilled from near the summit of Alto dell'Ortles glacier in the Italian Alps. The MS is generally well written and argues convincingly for the suggested age scale that is based on radio- metric ages and nuclear fallout products. The deepest meters of the ice cores contain ice that is more than 1000 years old, and the MS argues that the glacier was formed during the Northern Hemisphere Climatic Optimum

10  (NHCO). I have a few suggestions for the authors to consider:

In Figure 5, the orientation of the coordinate system is unclear to me. Are the X and ⌗Y directions following the GPR lines in Figure 1c or are they respectively along and perpendicular to the ice flow as stated in the main text? The best would probably be to show the orientation of the X-profiles and Y-profiles in a map. Are the units on the abscissas of Figure 5 meters?

Authors: The orientations of the X-profiles and Y-profiles follow the GPS lines and are now displayed in Fig. 6. The main text has been corrected accordingly. The units on the abscissas of Figure 5 are meters (added).

Referee #1: In section 6.2, the authors argue that the only way the obtained age profile can come about is if the glacier flow

20  pattern has recently changed significantly. I have difficulties following the argumentation of this section as not many details are given. A simple model is applied, but no results are presented.

Authors: We now report the simple model results in the new Figure 12.

Referee #1:  There is one observation in Figure 5 that seems important to me in this context. The inclination of the 45 m deep melt layer seen in the GPR profile and in the cores is very steep. This isochrone suggests that the oldest ice of the glacier probably is to be found in right hand side of figures 5 (X-profiles), which is under the ice divide, if I got the geometry right. Alternatively, it suggests that the oldest ice is to be found in higher depth resolution below the ice divide. I'm uncertain

30  if the authors take this observation into account, but to me the steep inclination of the melt layer suggests a significant increase in snow accumulation the further one gets away from the ice divide. Could it be that snow is blown away from the ice ridge and (re-)deposited further down the slope (on the lee side?). An increasing accumulation away from the ice ridge would probably lead to strong inclination of the deeper layers of the glacier and possibly explain why old ice is preserved close to the ice ridge where accumulation may be very low?

Authors: Referee #1 is correct. We observed modern snow accumulation at 3830 m, ~30 m below the drilling site (3859 m) and we determined a value of ~ 1000 mm w.e. (Festi et al. 2016), which is ~ 200 mm w.e. more than at the drilling site (3859 m). The point made by Referee #1 is now reported within the text (7-23).

Referee #1: In section 3, it is mentioned that limestone rock particles and pebbles are observed in the lowest meter of the ice cores. At the same time, the oldest ice is found in stratigraphic order in the same lowest meter of the cores. Indeed, the proposed age scale does look convincing based on the obtained C-14 ages, but still I am wondering how those pebbles got entrained in the ice if the ice is not disturbed (folded)?

Authors: Referee #1 is correct as the basal layers appear in stratigraphic order. In addition, the three stables isotopes profiles from the different cores match remarkably well down to the deepest ice layers (Fig. 10). One possibility is that, as already mentioned within the text, given the close rock outcrops near the Mt. Ortles summit, some relatively large pebbles were not entrained in the ice from the bedrock but from the glacier surface.

Referee #1: In section 2.2.2 it is suggested that the glacier bed could be lubricated by summer meltwater. This scenario seems rather implausible to me. If the ice is -2.8 C at 75 m depth, it seems highly unlikely that there is summer melt. There is no seasonal temperature variability possible at this depth.

20 Authors: The Mt. Ortles ice cores and the instrumental temperature record provide evidence of extensive surface melting on the Alto dell'Ortles summit during recent summers. Melt water percolation through the intersections of the glacier surface and bedrock, fractures and terminal crevasses, can thus lubricate the interface between basal ice and bedrock. There are several evidences in Svalbard and Greenland that seasonal meltwater can reach bedrock and change the ice velocity of cold-based ice caps and ice sheets (e.g. Dunse et al. The Cryosphere 2015; Bartholomew et al. Nature Geosc. 2010). We have
25 added this note to the text (14-7).

Bartholomew, I., Nienow, P., Mair, D., Hubbard, A., King, M. A., and Sole, A.: Seasonal evolution of subglacial drainage and acceleration in a Greenland outlet glacier, Nature Geosci., 3, 408-411, 2010.

30 Dunse, T., Schellenberger, T., Hagen, J. O., Kääb, A., Schuler, T. V., and Reijmer, C. H.: Glacier-surge mechanisms promoted by a hydro-thermodynamic feedback to summer melt, The Cryosphere, 9, 197-215, 10.5194/tc-9-197-2015, 2015.

Referee #1: In section 2.3 it is mentioned about the bedrock step that 'this feature does not completely enclose the drilling site in every direction'. In fact, the bump is only to one side of the cores, so I would suggest a reformulation.

Authors: Done. We now report that this feature is only on one side of the drilling site (7-30).

Referee #1: In Figure 3 it is hard to see details around the drilling sites in the main image. There is a white spot right below what appears to be BH3 and BH4. Is this a col, or does the spot indicate something else?

Authors: As mentioned in the caption of Fig. 3, the contour lines of the drilling dome are visible as the surface topography of the drill site was obtained from a LiDAR survey conducted exactly during the 2011 drilling campaign.

Referee #1: The authors argue that the ice started to form during the Holocene climatic optimum and that this same optimum is observed in the isotopic signature of nearby stalagmites. Is the NHCO seen in the isotopic signature of the deepest ice core ice? If the high frequency signal is removed? I cannot judge this from Figure 10.

Authors: Yes. Except the modern most enriched values, the lowest part of the ice core records, corresponding the NHCO, shows the most isotopically enriched values during the early-mid Holocene. This is one of the topics of another manuscript in preparation.

Referee #1: In Tables 2 and 3 it is difficult to see which samples correspond to each other as depth is transferred between the cores. Would it be possible to assign a unique name to each sample, so the reader can trace them from one table to the other?

Authors: We have improved Tables 2 and 3 in order to facilitate tracing of the same samples.

The conclusion seems to state the main findings in a bit disorganized way.

Authors: We have now subdivided this paragraph in two sections linked to the paleoclimatological and glaciological conclusions, respectively (16-7).

In point 6) is says the accumulation at the drilling site is 850 m/year. Probably this should be 850 mm/year?

Authors: corrected (16-26). Many thanks.

M. Kuhn

Authors: Thank you very much for your review and your comments.

5  M. Kuhn:  The glacier Alto dell' Ortles is an exceptional site for drilling into the Holocene history in the Eastern Alps. Its value and uniqueness were recognized in the Ortles Project (www.ortles.org) which produced three ice cores at 3859 m elevation that go back to about 7000 years and were analyzed by a large team of authors and laboratories who ensured the competent representation of the various disciplines involved in paleoclimate research. Analysis of ice cores requires them to be cold or polythermal, to have minimum annual accumulation and to have a flat topography that reduces the outflow of ice

10  and conserves it in place for millennia. These conditions are not met by many mountains in the Eastern Alps. Suter and others (2001) modelled the firn temperature in the Alps and found from altitude and exposition that cold firn should exist above 3400 m in northerly aspect and above 4150 on south slopes.

Authors: we have added this last sentence and have provided more emphasis to the Suter et al. 2001 reference in the

15  introduction (4-28).

In the Eastern Alps this would include the peaks of Disgrazia, Bernina, Adamello, Ortles, Cevedale, Ötztal, Stubai, Zillertal, Venediger and Goßglockner. The second condition, that there should be flat glacier tops, ruled out most of these mountains. Topographically suitable candidates like Adamello at 3539 or Weißseespitze at 3510 m in the Ötztal Alps, however, turned

20  temperate in the 1990s. Other peaks like Wildspitze 3768 m, and Venediger 3666 m may still be cold but their steep slopes do not keep ice in place for long. Alto dell' Ortles was timely chosen as research site. The lowermost meters of the Ortles ice cores are difficult to interpret, as is the slope of the internal layers detected by ground penetrating radar. They may reflect wind drift of snow, ablation or ice flow, all of which may have individual histories on a centennial time scale and may act on a very local scale as indicated by the bedrock contours in Fig. 6. As the paper refers to the Tyrolean Iceman – he too was

25  found in a shallow bedrock depression, protected from shearing ice motion. But shear must have occurred above at a vertical distance comparable to the horizontal scale of bedrock roughness, and such shear could be one explanation of a possible hiatus in ice core layering.

Authors: we believe that while the depression may have facilitated the in situ retention of old bottom ice, it is unlikely to

30  have caused a complete dynamical entrapment and the consequent formation of fossil ice decoupled from the upper stratigraphic sequence, which is also implied by the ice flow observed near bedrock.

This paper is of high current interest and deserves publication with minor changes as suggested by two previous comments.

I recommend numbering the profiles in Fig. 5 and entering them into Fig. 6.

Authors: done.

5    On page 2 / line 2 write Geologist instead of Geologin.

Authors: Geologin is the name of the company.

2/4: ENEA instead of Enea.

Authors: done. Thank you.

5-8 At the end of the introduction we have added a sentence that introduces the methodological part developed within the manuscript. In this way we hope that the reader will better follow the necessary and detailed presentation of the results.

11-1 We have carefully reviewed paragraph 4.3 linked to the carbon 14 analysis. In particular a different correction of the blanks has been applied (after Uglietti et al. 2016, The Cryosphere, this issue) resulting in minor changes in the carbon 14 dates. A different method (simple average) to combine carbon 14 results of different parts from one single ice section has also been used (after Uglietti et al. 2016, The Cryosphere, this issue) resulting in minor changes of the calculated age.

10 Finally, we have decided to use the year 2012 as a reference for "present" as it better reflects the time when the drilling operations were conducted in October 2011 (2011.8).

12-24 Empirical fitting of the age model has been conducted using the mid-depth of the samples expressed in water equivalent. This will also allow evaluating the strain rate while it causes minor changes in the depth-age relationship.

15-28 We have added in the discussion the reference of Ilyashuk et al. 2001 that 
[revised manuscript text omitted]

Paolo Gabrielli 10/17/2016 12:17 PM

Jenk Theo Manuel 10/11/2016 9:48 AM

Paolo Gabrielli 10/7/2016 3:56 PM

Jenk Theo Manuel 10/11/2016 9:47 AM

Jenk Theo Manuel 10/11/2016 11:04 AM

Jenk Theo Manuel 10/11/2016 11:05 AM

Jenk Theo Manuel 10/11/2016 10:52 AM

Paolo Gabrielli 10/13/2016 4:30 PM

Jenk Theo Manuel 10/11/2016 10:55 AM

Paolo Gabrielli 10/17/2016 12:22 PM

Jenk Theo Manuel 10/11/2016 11:10 AM

Paolo Gabrielli 10/13/2016 3:55 PM

Jenk Theo Manuel 10/11/2016 10:51 AM

Paolo Gabrielli 10/7/2016 3:58 PM

Jenk Theo Manuel 10/11/2016 11:14 AM

Paolo Gabrielli 10/7/2016 3:59 PM

Jenk Theo Manuel 10/11/2016 11:16 AM

Paolo Gabrielli 10/7/2016 4:01 PM

Jenk Theo Manuel 10/11/2016 11:17 AM

Paolo Gabrielli 10/7/2016 4:03 PM

Paolo Gabrielli 10/7/2016 4:04 PM

[revised manuscript text omitted]

*All depths referred to the Ortles core #2 depth scale

5   **Table 3: Data used in the depth-age modelling.**